# Effects of Biochar on the Fluorescence Spectra of Water-Soluble Organic Matter in Black Soil Profile after Application for Six Years

**DOI:** 10.3390/plants12040831

**Published:** 2023-02-13

**Authors:** Liang Jin, Dan Wei, Yan Li, Guoyuan Zou, Lei Wang, Jianli Ding, Yitao Zhang, Lei Sun, Wei Wang, Xingzhu Ma, Huibo Shen, Yuxian Wang, Junqiang Wang, Xinrui Lu, Yu Sun, Xinying Ding, Dahao Li, Dawei Yin

**Affiliations:** 1Plant Nutrition and Resources Institute, Beijing Academy of Agriculture and Forestry Sciences, Beijing 100097, China; 2Institute of Geographic Sciences and Natural Resources Research, Beijing 100101, China; 3Heilongjiang Institute of Black Soil Protection and Utilization, Harbin 150086, China; 4Qiqihar Branch of Heilongjiang Academy of Agricultural Sciences, Qiqihar 161005, China; 5Northeast Institute of Geography and Agroecology, Chinese Academy of Sciences, Changchun 130012, China; 6Institute of Crop Cultivation and Tillage, Heilongjiang Academy of Agricultural Sciences, Harbin 150027, China; 7Institute of Animal Husbandry and Veterinary of Heilongjiang Academy of Agricultural Sciences, Qiqihar 161005, China; 8Qiqihar Agricultural Technology Promotion Center, Qiqihar 161000, China; 9College of Agricultural Science and Technology, Heilongjiang Bayi Agricultural University, Daqing 163319, China

**Keywords:** biochar, black soil region, water solubility, fluorescent component, vertical depth of soil

## Abstract

At present, extracting water-soluble organic matter (WSOM) from agricultural organic waste is primarily used to evaluate soil organic matter content in farmland. However, only a few studies have focused on its vertical behavior in the soil profile. This study aims to clarify the three-dimensional fluorescence spectrum characteristics of the WSOM samples in 0–60 cm black soil profile before and after different chemical fertilizer treatments after six years of fertilization. Fluorescence spectroscopy combined with fluorescence and ultraviolet-visible (UV-Vis) spectroscopies are used to divide four different fertilization types: no fertilization (T0), nitrogen phosphorus potassium (NPK) (T1), biochar (T2), biochar + NPK (T3), and biochar + N (T4) in a typical black soil area. The vertical characteristics of WSOC are also analyzed. The results showed that after six years of nitrogen application, T2 had a significant effect on the fluorescence intensity of Zone II (decreasing by 9.6% in the 0–20 cm soil layer) and Zone V (increasing by 8.5% in the 0–20 cm soil layer). The fluorescent components identified in each treatment group include ultraviolet radiation A humic acid-like substances (C1), ultraviolet radiation C humic acid-like substances (C2), and tryptophan-like substance (C3). As compared with the land with T1, the content of C2 in the 20–60 cm soil layer with T2 was lower, while that of C2 in the surface and subsoil with T3 was higher. In addiiton, there were no significant differences in the contents of C1, C2, and C3 by comparing the soils applied with T3 and T4, respectively. The composition of soil WSOM was found to be significantly influenced by the addition of a mixture of biochar and chemical fertilizers. The addition of biochar alone exerted a positive effect on the humification process in the surface soil (0–10 cm). NPK treatment could stimulate biological activity by increasing biological index values in deeper soil layers (40–50 cm). Nitrogen is the sovereign factor that improves the synergism effect of chemical fertilizer and biochar during the humification process. According to the UV-Vis spectrum and optical index, soil WSOM originates from land and microorganisms. This study reveals the dynamics of WSOC in the 0–60 cm soil layer and the biogeochemical effect of BC fertilizer treatment on the agricultural soil ecosystem.

## 1. Introduction

Water-soluble organic matter (WSOM) is often regarded as one of the most critical components for crops because it can be decomposed by microbes, and thus, releases nutrients for plant uptake. As a result of intensive agricultural operations, the input of fertilizers into agricultural farmland, especially organic materials, has been extensively studied. It has been reported that WSOM is the most active organic carbon pool in the terrestrial ecosystem and a significant contributor to total carbon accumulation in the soil, as the reduction of water-soluble organic carbon (WSOC) is mainly due to abiotic reaction rather than mineralization [1,2]. A soil’s natural fertility is often dependent on the quality and quantity of soil organic carbon (SOC). Given its crucial role in soil fertilization processes, WSOM can serve as a significant mobile component of SOC, and therefore, it has been studied extensively [3,4,5]. Moreover, WSOM is commonly incorporated in agricultural farmlands for maintaining soil structure and improving carbon sequestration. Recently, many studies have shifted focus to WSOM with organic amendments. Most biological waste materials of plant origin contain a large amount of dissolved organic matter (DOM). Amending plant residues and some organic fertilizers (including poultry manure) can increase the pH value and enhance the solubility of soluble organic matter (SOM). An increase in DOM after organic amendment input might be momentary. Long-term field trials have been beneficial to further understanding the synthesis of sustainable organic matter management on a net increase in DOM as compared with conventional agriculture [6,7].

Biochar (BC) has attracted extensive attention as a porous organic soil amendment since WSOM products can improve soil aging. Fan et al. [8] demonstrated that BC treatment increased oxygen-containing functional groups and enhanced the complexation capability of the WSOM. Their results confirmed that significant numbers of aromatics were released from the BC into the soil during the aging process, thus, increasing the dissolved organic carbon (DOC) and the aromaticity of WSOM. BC has been effectively applied in leaching SOM. The properties of SOM can be diverse for various biomass sources and pyrolysis conditions. Ten BC samples produced from different feedstocks were characterized using emission electron microscope/parallel factors (EEM/PARAFAC) to evaluate the compositional features of DOM. The results showed that the abundance of the protein-/tannin-like component in the WEOM phase of BC was influenced mainly by feedstock [9]. The performance of WSOM extract of wheat-straw-derived BC on Chinese cabbage growth promotion depended on pyrolysis temperature, which was more outstanding at 350 °C than at 450 °C and 550 °C [10]. The optical properties of DOM were more dependent on the BC types with different origins (sludge, corn, and rice) rather than on the extraction time [11]. BC decreased the DOC from the soil in proportion to the additional rates, and BC from different feedstock amendment increased the aromaticity levels of the earth DOM [12]. BC has previously been observed to possess selective sorption of DOM [13], which requires simultaneous observation of DOM content and composition. Some scholars have studied the content and characteristics of soil WSOM at the concentrations of 0, 20, and 40 t/ha after returning straw BC to the field for eight years. The results showed that BC-leaching WSOM had high aromaticity, and BC returning could increase oxygen-containing functional groups and could enhance the complexation ability of WSOM.

At present, mixing different proportions of BC with chemical fertilizers has been widely used in soil to change the soil WSOM. BC of 20 or 30 t/ha has been shown to increase soil carbon content as compared with an untreated control group [14]^.^ Previous research has found that the addition of BC (different addition rates of 4 t/ha and 8 t/ha) significantly influenced the content of soil DOM. In addition, under the same fertilization conditions, a large amount of BC addition dramatically increased the average concentration of soil DOC in a wheat-maize rotation system (from 83.99 mg/kg to 144.27 mg/kg), while maintaining the composition of DOM [15]. Carvalho et al. [16] investigated the effects of a single application of wood BC combined with inorganic nitrogen fertilizer on soil chemical and physical properties, and found that the fraction of oxidized carbon in soil increased with increased BC application. The high aromaticity of WSOM was evidenced in a recent study of long-term BC amendment of soil [8]. BC-derived DOM is primarily composed of aliphatic compounds, condensed aromatics, phenolics, and polyphenols [17], depending on the BC feedstock, production conditions, and environmental conditions.

Under the background of soil BC improvement [18], in this paper, we focus on the effect of fertilization conditions on fertile black soil and strive to provide practical suggestions for soil improvement in black soil areas. A high proportion of BC applications has previously been considered to be impractical or economically infeasible [19]. There have been many studies on BC and N application rates [20]. Some researchers have analyzed the composition and structure of DOM in various BC feedstocks, straw BC produced at different pyrolysis temperatures, as well as the effects of fresh and aged BC on the behavior of herbicide mesotrione and the effect of BC on accumulation characteristics of DOM in substrate soil [21,22,23,24,25,26,27,28,29].

Still, only a few efforts have been made on the structure and characteristics of WSOC in the vertical profile of fertile soil under the low application rate of BC [30]. Therefore, in this paper, we innovatively explore the effect of low-dose BC (3.6 t/ha, in situ corn straw biomass equivalent BC) on a black soil field. The vertical characteristics of WSOM in typical black soil under different organic matter inputs (no fertilization, chemical fertilizer, BC, BC + chemical fertilizer, and BC + nitrogen fertilizer, which refers to T0 (CK/no fertilizer), T1 (NPK), T2 (BC), T3 (NPK + BC), T4 (N + BC), respectively) after six years of fertilization are evaluated by using three-dimentional (3D) fluorescence spectroscopy. The specific objectives are as follows: (1) to study the fluorescence characteristics of WSOM in a two-dimentional (2D) vertical soil profile; (2) to evaluate the spectral characteristics of WSOC in the soil profile after application of BC and other fertilizers (3D); and (3) to assess the correlations among WSOC parameters. Spectral characteristics and fluorescence characteristics refer to the fluorescence spectral characteristics of substances, which represent the structural composition and distribution of WSOC in this research.

## 2. Materials and Methods

### 2.1. Site Description

As one of the largest mollisol regions in the world, the black soil region of northeast China has the most fertile soils, with excellent chemical and physical characteristics [31]. However, this region now undergoes severe soil degradation due to excessive land reclamation. Therefore, black soil degeneration has become a bottleneck for the sustainable production of grains in this region. This long-term study was conducted in Minzhu Town, Daowai District, Haerbin City, China (N 45°50′3″, E 126°51′05″). The meteorological data indicated that the long-term average annual rainfall in this area ranged from 486.4 to 543.6 mm. Approximately 80% of precipitation occurred from June to September, and the soil was classified as Chernozem [32]. The elevation was 138 m above the mean sea level, and the depth to the groundwater table was 80 m. The annual average wind speed was 4.1 m/s, and the maximum speed was 18.9 m/s. The reference soil data used here were obtained from the Northeastern Asia Geographic Scientific Data Center (http://www.igadc.cn/nearests/u5b1a, accessed on 22 Feburary 2021).

### 2.2. Design Patterns

The field experiment was carried out with soybean/maize rotation from 2013 (Soil basic properties in Table 1) to 2018. The main treatments consisted of five fertilization types: (T0i) CK (control check), (T1ii) NPK, (T2iii) 3.6 t/ha BC (3.6 t/ha), (T3iv) NPK + 3.6 t/ha BC (3.6 t/ha), and (T4v) N + BC (3.6 t/ha) BC. All treatments were replicated three times. The formula for calculating the BC application rate was as follows: 3.6 t/ha (BC, application per year) = 10.5 t/ha (maize straw biomass production per year) × 35% (rate of biomass conversion into BC). Soybean received subsurface-applied preplant fertilizer of 68 kg K_2_O/ha, 47 kg N/ha, and 78 kg P_2_O_5_/ha. Spring maize received 105 kg KCl/ha, 176 kg N/ha, and 77 kg P_2_O_5_/ha. Spring soybean cultivar Heinong 58 was sowed on 14 May 2013 and harvested on 8 October 2013 [33]; while Spring maize cultivar Longdan 42 was sowed on 1 May 2014 and harvested on 8 October 2014. From 2015 to 2018, soybean and maize were planted in turn. The NPK fertilizer for soybean was applied preplant, whereas the N fertilizer for maize was applied as urea at a rate of 176 kg N/ha as the base fertilizer and the remaining 50% at the jointing stage. As a soil conditioner, BC (Table 2) was sprinkled on the ridge next to the ridge every year from 2013 to 2018 It was thoroughly mixed with the soil with a tiller and plowed to a depth of at least 20 cm. Soil samples were collected in 2018 after harvesting by an earth auger (diameter 10 cm) based on the 60 cm soil profile. Briefly, the soil was sampled from the 0–60 cm soil layers at 10 cm intervals, followed by air-drying for one week at room temperature. After sieving through a 2 mm mesh, the soil samples were kept at 4 °C until further analyses.

### 2.3. Measurements

#### 2.3.1. DOM Extraction

Water-soil oscillation was performed to extract the soil DOM. Briefly, each soil sample was mixed with deionized water at a solid-to-water ratio of 1:6 (*w/v*) and incubated for 24 h with continuous shaking (180 rpm). After centrifugation at 10,000× *g* for six minutes, the suspension was filtered through a cellulose acetate membrane filter (pore size 0.45 μm). After filtration, the samples were kept at −20 °C for the three-dimensional fluorescence spectral analysis. The organic carbon levels of the filtrate were detected using a total organic carbon (TOC) analyzer (TOC-VCPH, Shimadzu, Japan) [34]. Each measurement was carried out in triplicate.

#### 2.3.2. Ultraviolet-Visible (UV-Vis) Spectroscopy and Fluorescence Spectroscopy

DOM is often quantified by its carbon content, represented as DOC or TOC [35]. In this study, the levels of TOC were detected using a TOC analyzer (TOC-L, Shimadzu, Japan). UV-Vis measurements were performed at 250–400 nm using a UV spectrophotometer (UV-1780, Shimadzu, Japan) coupled with a 10 mm quartz cuvette. The scanning speed and interval were medium and 1 nm, respectively. The UV-Vis spectral parameters adopted in this experiment included specific ultraviolet light absorbance at 254 nm (SUVA254) and spectral recording (SR).

During the construction of excitation–emission matrices (EEMs), the fluorescence intensities were recorded using a fluorescence spectrometer (RF-6000, Shimadzu, Japan) at the emission and excitation wavelengths of 250–550 and 200–500 nm, respectively. The scanning interval and bandwidth were 2 and 3 nm for emission wavelength, while both were 5 nm for excitation wavelength. The scanning speed was maintained at 6000 mm/min, and the inner filter effects were corrected through blank subtraction (Milli-Q water). The measurement indices used here were tje biological index (BIX), fluorescence index (FIX), and humification index (HIX) [36,37].

#### 2.3.3. Parallel Factors (PARAFAC) Modeling

To characterize the fluorescence components of DOM, the PARAFAC model was constructed using MATLAB 7.0 (Mathworks, Natick, MA, USA) with the DOM Fluor toolbox. The correct number of components in the model was measured according to the core consistency diagnostic and split-half validation tests. The difference among various elements in each sample was further compared by estimating the relative contribution of each component to the total EEM.

#### 2.3.4. Statistical Analysis

The SPSS software ver. 20.0 (SPSS Inc., Chicago, IL, USA) was employed for data analysis. After checking homogeneity and normality, the data with homogenous variances and normal distribution were analyzed using the one-way analysis of variance and least significant difference (LSD) test to compare the WSOC contents in various land-use types and soil depths. Meanwhile, the correlations among different WSOM parameters were determined by Pearson’s correlation coefficient (r). A *p*-value of 0.05 or even 0.01 was deemed to be statistically significant. All data were expressed as means ± standard deviations.

## 3. Results

### 3.1. The Fluorescence Properties (2D) of WSOM at Different Soil Depths

Figure 1 displays the three-dimensional fluorescence spectral data of WSOC in the soil profile after six years of fertilization. Table 1 presents the original data of the results in Figure 1. Murphy et al. [38] divided a spectrograph into five sections according to the emission and excitation wavelengths of target molecules: Section I (250–330 nm/200–250 nm) containing tyrosine-like substance, Section II (330–380 nm/200–250 nm) containing tryptophan-like substance, Section III (380–550 nm/220–250 nm) containing fulvic acid-like substance, Section IV (250–380 nm/250–600 nm) containing soluble microbial by product-like substance, and Section V (380–600 nm/250–600 nm) containing the humic acid-like substance.

As shown in Figure 1, the fluorescence intensities and spectrogram’s shape varied with the soil depths. The WSOC of all treatment groups consisted of tyrosine, tryptophan, fulvic acid, soluble microbial by-products, and humic acids. After six years of fertilization, Sections III and V achieved the maximum fluorescence intensities (over 60% for Section V). In contrast, as shown in Table 3, as compared with the NPK treatment group (T1), the BC treatment group (T2) could significantly affect the fluorescence intensities of Section II (decreased by 9.6% in the 0–20 cm layer) and Section IV (increased by 8.5% in the 0–20 cm layer). This phenomenon suggests that the accumulation of aromatic substances leads to a reduction in tryptophan substances and enrichment of humic acids. Furthermore, the BC + NPK treatment (T3) remarkably increased the fluorescence intensities in Section IV (soluble microbial byproduct-like substance, by 5.9% in the 0–20 cm layer) and dramatically reduced those in Section I (tyrosine-like substance, by 26.1% and 13.7% in the 0–20 and 20–30 cm layers, respectively), which was in agreement with the findings that an increase in soluble microbial byproduct-like materials significantly affect organic nitrogen transformation [39].

### 3.2. Excitation–Emission Matrices (EEMs) and Fluorescence Characteristics (3D) of DOM

The EEMs (Figure 2) and compound identities of Components 1–3 (C1–3) are summarized in Table 4. For all treatment groups except for T1, the peak of C1 was detected at 406 nm emission and 235 nm excitation wavelengths, which could be attributed to UVA humic acid-like structures. This compound is recognized as a low-molecular-weight humic acid that has been commonly found in agricultural and wetland soils [40].

In addition, the peak of C2 was detected at 462 nm emission and 260 nm excitation wavelengths, which could be ascribed to a high-molecular-weight UVC humic acid-like substance [45]. Previous research has shown that the length of emission wavelength can determine the number of aromatic and conjugated humic-like substances. Furthermore, the peak of C3 was detected at 342 nm emission and 225 nm excitation wavelengths, which corresponded to a tryptophan-like molecular compound [46,47].

### 3.3. Identification of WSOC Components and Their Correlations with Each Other

The relative abundances of C1–3 in all treatment groups, except for the T2 group, showed that the levels of UVC humic-like substance (C2) were decreased, and those of protein-like substance (C3) were increased with increasing soil depths. Moreover, the levels of UVA humic acid-like substance (C1) were higher than those of UVC humic-like substance (C2), while the protein-like substance had minor abundance in all treatment groups. An opposite trend was observed for the levels of tryptophan-like substance (C3), and those of C2 remained over 50% in deeper soil layers (20–60 cm). In addition, the levels of humic acid-like substances (C1 and C2) in the subsoil layer (20–60 cm) were reduced by approximately 5% in BC (T2) as compared with those in NPK (T1) after application for six years (Figure 3a). Moreover, the NPK + BC treatment showed the lowest content of C1 and the highest content of C2 distributed in both topsoil and subsoil layers. However, there were no significant differences in the three components (C1, C2, and C3) between T3 and T4 groups in topsoil and subsoil layers (0–20 cm and 20–60 cm, respectively), as shown in Figure 3. The corresponding reason is that nitrogen and its transformation are the primary essences in regulating the process of humification. Table 5 provides the original data of the results in Figure 3.

Several optical indices were used to evaluate the geographical origin and quality of WSOM, including FIX, HIX, and BIX (Figure 4). It was found that the FIX values of different treatment groups remained over 1.59 (mean value). The lowest FIX value (1.52) was observed in the topsoil layer (0–10 cm) under BC application, while the highest value was detected in the subsoil layer (40–50 cm) under the combination of BC and NPK fertilizer (Figure 4a).

The HIX values were around 0.27 (mean value) among different treatment groups. As compared with the decreased HIX values with increasing soil depths in the control (T1) group, the mean HIX values of BC-amended soil in different soil layers were higher as compared with those of NPK-fertilized soil. As expected, the HIX values of the T3 treatment group were higher than those of the T2 treatment group, which were consistent with previous results that higher HIX values were associated with the presence of more complex molecules containing aromatic rings or compounds with high molecular weight [38,42].

Additionally, the average HIX values of WSOM in both topsoil and subsoil layers were lower in the T3 group than those in T4 group, indicating that NPK plays a more important role than nitrogen during humification. Furthermore, the BIX values of all treatment groups exhibited the same increasing trends. Notably, the mean values of BIX distributed in soil profiles were within the range of 0.63–0.67. As shown in Table 6, BIX demonstrated highly significant positive correlations with FIX and C3 (*p* < 0.01).

## 4. Discussion

The combination of chemical fertilizer and BC shift the composition of WSOM distribution in 0–60 cm depth soil. The results from Figure 2 indicate that humic acid-like and fulvic acid-like organic matters are the most dominant in black soil with and without amendment [48]. As compared with the BC (T2) group, the BC + NPK (T3) group significantly increased the fluorescence intensities in Section II (by 13.3% in the 0–20 cm soil layer), Section III (by 8.4% in the 0–20 cm soil layer), and Section IV (by 50.1% in the 0–20 cm soil layer). Therefore, the enrichment of tryptophan-like substance, fulvic acid-like substance, and soluble microbial byproduct-like substance is influenced by the synergistic effect of BC plus NPK fertilizer. Meanwhile, treatment with BC + N fertilizer (T4) exhibited increasing trends in Section I (by 20.7%), Section II (by 12.2%), and Section IV (by 28.4%) for the topsoil layer (0–20 cm). It demonstrates a positive, cooperative effect between the organic–inorganic fertilizer mixture and topsoil layer, mainly due to its aeration condition, thereby, improving the production of soluble microbial byproduct-like substance.

It can be seen that the primary components of WSOM were humic acid-like substances (C1 and C2) and protein-like substances (C3), which were consistent with those reported in previous studies [49,50]. Under BC treatment (T1), C2 was identified as an aquatic humic acid, which was consistent with previous findings [51]. Moreover, it has been suggested that the content of WSOM was elevated with increasing soil depths, possibly via microbial biodegradation [1]. These observations (Figure 4) imply that the WSOM in the topsoil layer (0–20 cm) consists of a humic acid-like substance. The WSOM in subsoil layers (20–60 cm) is generated from microbial decomposition, and the levels of microbial by-products are elevated with increasing soil depths [52], which is consistent with the evidence that the products of microbial processing of WSOM can accumulate aromatic in the subsoil [1]. However, the distribution of WSOC in the T2 treatment group exhibited the opposite trend in the two soil layers, which could be affected by the anti-degradation properties of BC. The concentrations of water-soluble organic carbon (WSOC) in the BC-only amended soils were significantly lower than those in the control soils, indicating that BC decreased the leaching of DOM [12]^.^ In most treatment groups, C2 could be ascribed to an aromatic high-molecular-weight terrestrial humic acid broadly distributed in the forest and wetland soils [44]. By comparing the findings of previous studies [53,54] with ours, it can be inferred that the WSOM of maize straw-derived BC input contains similar terrestrial and autochthonous/allochthonous integration sources in the topsoil layer (0–20 cm) with moderate neogenesis characteristics and exhibits strong authigenic characteristics in the subsoil layer (20–60 cm).

In addition, the relative abundances of the three components were estimated by PARAFAC modeling to further understand the changes in soil WSOM after six years of BC and chemical fertilizer application. The results are shown in Figure 4. Optical indices including the humification index, fluorescence index, and biological index can provide information about the general structure and the chemical characteristics of soil DOM [55]. In this study, three indexes were calculated and are shown in Figure 4. For HIX, the difference was observed in the following treatment groups: The value of the BC treatment group was obviously higher as compared with other treatment groups in the top soil (0–10 cm), which indicated that the application of BC alone might favor the humification process due to its specific characteristics. Sanchez-Monedero et al. [56] also found that BC could improve the humification process by adsorption onto the surface of easy-degradable compounds and aromatic moieties. Nielsen et al. [57] observed that the BIX fixation peak of an NPK fertilizer group in soil at a depth of 50 cm was related to an increase in NO_3_^−^. Accumulation of NO_3_^−^ may stimulate the activity of microorganisms at the same BIX peak and soil depth without BC application. The lowest FIX value was found in the surface soil (0–10 cm) after BC treatment, possibly due to an increase in C/N ratio and accumulation of stable BC particles that delayed the humification process [58].^.^ As reported by Gao et al. [23], there was no significant relationship between FIX and C1. Furthermore, similar to other studies [59,60], this study demonstrated that the FIX value was independent of WSOC concentration. Nevertheless, further studies on the changes of humic acid and fulvic acid in soil profiles are needed to fully reveal DOM’s composition and transformation mechanism over the long-term amendment of BC for soil remediation. Significant changes in the existing indicators demonstrate that the soil WSOM composition changes significantly after six years of BC and chemical fertilizer application.

## 5. Conclusions

EEM-PARAFAC was applied to evaluate the influence of BC amendments on WSOM characteristics in soil profiles through a long-term field experimental soybean-maize rotation system in the typical black soil region of Northeast China. The composition of soil WSOM was found to be significantly influenced by the addition of a mixture of BC and chemical fertilizers after six-year amendment. The abundances of tryptophan-like, fulvic acid-like, and soluble microbial byproduct-like substances were significantly enriched by the synergistic effect of BC plus NPK fertilizer. The addition of BC alone exerted a positive effect on the humification process in the surface soil (0–10 cm). The NPK treatment could stimulate biological activity by increasing BIX values in deeper soil layer (40–50 cm).

Three identified fluorescent components (UVA humic-like substance, UVC humic-like substance, and tryptophan-like substance, referred to as C1, C2, and C3) were identified. As compared with T1, humic acid-like substances (C1 and C2) in the subsoil layer (20–60 cm) were decreased by about 5% in T2 after application for six years. The fluorescence intensities of these three components (especially humic-like material) were enhanced with increasing soluble microbial byproduct-like substance levels in both BC and NPK treatment groups in the top soil layers (0–20 cm), but the composition of WSOM in the T3 and T4 groups had no significant differences. Nitrogen is the sovereign factor that improves the synergism effect of chemical fertilizer and BC during the humification process. These findings would be beneficial to comprehend the effects and processes of combined BC and chemical fertilizers on the spectral and fluorescence characteristics of WSOM from black soil profiles.

## Figures and Tables

**Figure 1 plants-12-00831-f001:**
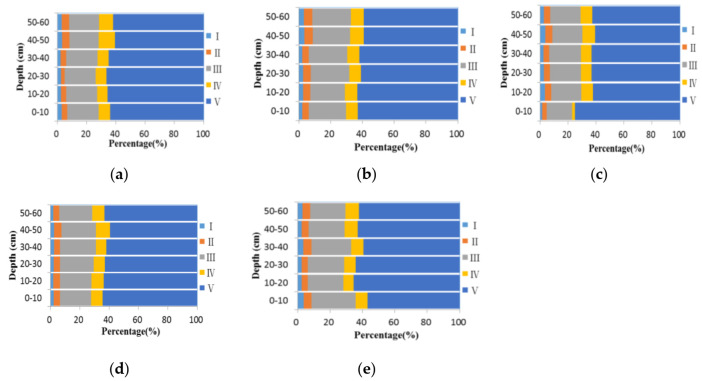
Fluorescence characteristics of WSOC after six years of fertilization (I, tyrosine-like substance; II, tryptophan-like substance; III, fulvic acid-like substance; IV, soluble microbial byproduct-like substance; V, humic acid-like substance): (**a**) no fertilizer; (**b**) NPK; (**c**) BC; (**d**) NPK + BC; (**e**) N + BC.

**Figure 2 plants-12-00831-f002:**
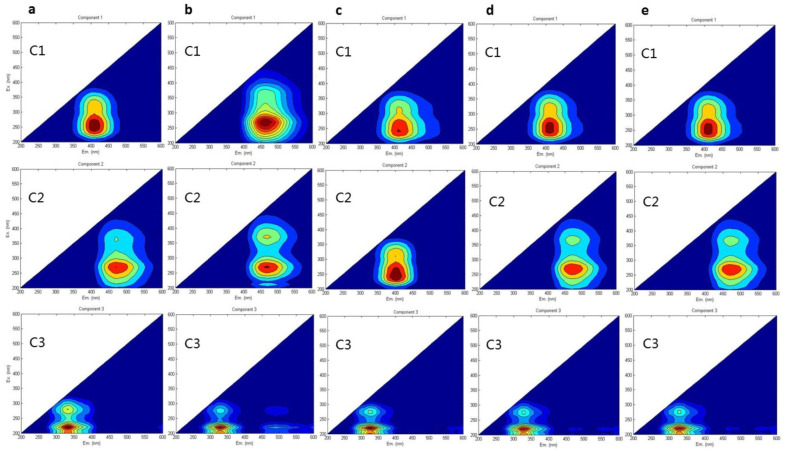
EEMs of the five treatment groups: (**a**) CK; (**b**) NPK; (**c**) BC; (**d**) NPK + BC; and (**e**) N + BC.

**Figure 3 plants-12-00831-f003:**
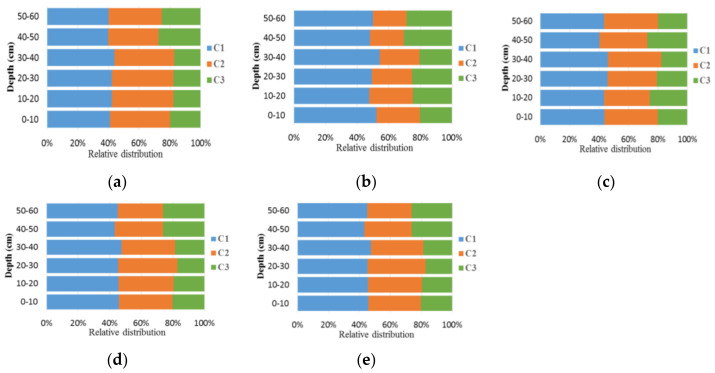
Relative abundances of the three fluorescence components (C1, UVA humic-like substance; C2, UVC humic-like substance; and C3, tyrosine-like (protein-like) substance) estimated by PARAFAC modeling in 0–60 cm soil layers with different fertilizer input for six years: (**a**) CK; (**b**) NPK; (**c**) BC; (**d**) NPK + BC; (**e**) N + BC).

**Figure 4 plants-12-00831-f004:**
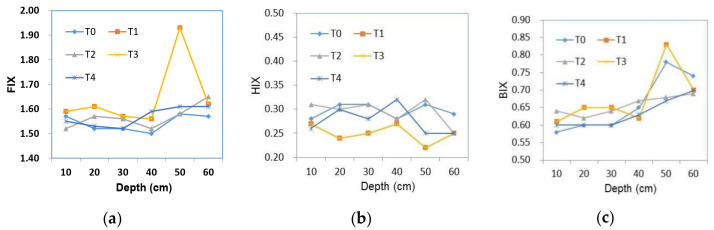
Fluorescence index (FIX), humification index (HIX), and biological index (BIX) of the five treatment groups in the whole soil depth (0–60 cm): (**a**) FIX; (**b**) HIX; (**c**) BIX.

**Table 1 plants-12-00831-t001:** Basic properties of the 0–60 cm soil layers.

Depth	Total N	Total P	Total K	Avail. * N	Avail. * P	Avail. * K	SOM **	pH	BD ***
cm	g/kg	g/kg	g/kg	mg/kg	mg/kg	mg/kg	g/kg		g/cm^3^
0–30	1.45	1.05	25.52	163.3	20.61	187.92	29.87	6.74	1.31
30–60	0.61	0.66	24.06	86.80	14.50	179.00	13.85	7.45	1.41

* Avail., available; ****** SOM, soil organic matter; ******* BD, bulk density.

**Table 2 plants-12-00831-t002:** Particle components of the BC.

Particle Components (%)											
SOC * g/kg	Ca g/kg	K g/kg	Mg g/kg	N g/kg	O g/kg	P g/kg	Si g/kg	pH	<0.1 mm	0.1–2 mm	>2 mm
598	3	17.0	2	7.85	166	1.327	60	8.69	15.0	60.2	24.8

* SOC, soil organic carbon.

**Table 3 plants-12-00831-t003:** The original data of the results in Figure 1 (%).

EX/EM		0–10 cm	10–20 cm	20–30 cm	30–40 cm	40–50 cm	50–60 cm
**CK**	**Ⅰ**	2.9	2.4	2.4	2.3	3.2	3.1
	**Ⅱ**	3.8	3.7	2.7	3.7	5.1	4.9
	**Ⅲ**	21.6	21.1	21.1	21.5	20.1	20.5
	**Ⅳ**	7.8	7.3	7.3	7.6	11.0	9.5
	**Ⅴ**	64.0	65.6	66.5	64.9	60.6	62.0
**NPK**	**Ⅰ**	2.21	2.77	2.72	2.23	3.55	3.34
	**Ⅱ**	4.05	4.58	4.83	4.19	5.34	5.19
	**Ⅲ**	23.45	21.60	24.28	24.17	23.42	24.18
	**Ⅳ**	7.27	7.79	7.36	7.44	8.54	8.21
	**Ⅴ**	63.02	63.26	60.82	61.97	59.15	59.08
**BC**	**Ⅰ**	1.9	3.0	1.0	4.5	4.8	4.0
	**Ⅱ**	3.6	4.6	4.5	5.8	5.6	6.3
	**Ⅲ**	22.2	23.4	23.2	23.1	22.9	22.9
	**Ⅳ**	7.0	7.5	6.4	7.6	7.8	7.5
	**Ⅴ**	65.3	61.6	64.9	59.0	59.0	59.2
**NPK + BC**	**Ⅰ**	1.53	3.11	3.63	2.18	2.75	2.39
	**Ⅱ**	5.62	5.61	5.36	3.95	4.76	4.21
	**Ⅲ**	21.09	22.83	23.39	23.56	22.92	21.08
	**Ⅳ**	6.51	6.68	6.96	6.31	7.89	7.46
	**Ⅴ**	65.25	61.77	60.66	64.00	61.68	64.87
**N** **+BC**	**Ⅰ**	3.01	2.40	0.74	2.14	3.19	3.21
	**Ⅱ**	4.42	3.67	3.43	3.74	4.55	4.77
	**Ⅲ**	25.81	22.80	21.75	21.88	21.77	21.66
	**Ⅳ**	9.17	6.93	6.58	7.46	8.33	7.16
	**Ⅴ**	57.59	64.21	67.50	64.78	62.15	63.19

**Table 4 plants-12-00831-t004:** EEM locations and the three fluorescence components determined by PARAFAC modeling in the five treatment groups.

Group	Approximate EEM Location			Fluorescence Compounds		
	C1	C2	C3	C1	C2	C3
**CK**	Em: 407 nm Ex: 256 nm	Em: 474 nm Ex: 263 nm	Em: 332 nm Ex: 221 nm	Ultraviolet A (UVA) fulvic acid-like (terrestrial humic substance)	Ultraviolet C (UVC) humic acid-like (terrestrial humic substance)	Tryptophan-like substance (protein components)
**NPK**	Em: 412 nm Ex: 243 nm	Em: 468 nm Ex: 270 nm	Em: 325 nm Ex: 220 nm	UVA fulvic-like acid (terrestrial humic substance)	UVC humic acid-like (terrestrial humic substance)	Tryptophan-like substance (protein components)
**BC**	Em: 461 nm Ex: 272 nm	Em: 399 nm Ex: 243 nm	Em: 324 nm Ex: 219 nm	UVC humic acid-like (terrestrial humic substance)	UVA fulvic acid-like (aquatic humic substance)	Tryptophan-like substance (protein components)
**NPK + BC**	Em: 410 nm Ex: 249 nm	Em: 467 nm Ex: 270 nm	Em: 326 nm Ex: 221 nm	UVA fulvic-like acid (terrestrial humic substance)	UVC humic acid-like (terrestrial humic substance)	Tryptophan-like substance (protein components)
**N + BC**	Em: 409 nm Ex: 245 nm	Em: 468.5 nm Ex: 269 nm	Em: 325.5 nm Ex: 220 nm	UVA fulvic-like acid (terrestrial humic substance)	UVC humic acid-like (terrestrial humic substance)	Tryptophan-like substance (protein components)

references [41,42,43,44].

**Table 5 plants-12-00831-t005:** The original data of the results in Figure 3 (%).

Depth (cm)		0–10	10–20	20–30	30–40	40–50	50–60
**CK**	C1	41.10	42.28	42.28	43.93	39.81	40.11
	C2	39.10	40.10	40.10	38.92	32.73	34.53
	C3	19.80	17.62	17.62	17.15	27.46	25.36
**NPK**	C1	52.16	47.60	49.31	54.22	48.20	49.92
	C2	27.51	27.71	25.31	25.35	21.24	21.15
	C3	20.32	24.69	25.38	20.43	30.56	28.93
**BC**	C1	45.79	45.60	45.41	47.38	43.27	45.02
	C2	33.85	34.77	37.32	34.10	30.58	28.86
	C3	20.36	19.62	17.27	18.52	26.15	26.12
**NPK** **+BC**	C1	45.70	45.50	45.32	47.29	43.16	44.91
	C2	33.96	34.88	37.41	34.20	30.70	28.98
	C3	20.34	19.62	17.27	18.51	26.14	26.11
**N** **+BC**	C1	45.70	45.51	45.33	47.29	43.18	44.93
	C2	33.96	34.88	37.42	34.20	30.70	28.98
	C3	20.34	19.61	17.26	18.50	26.13	26.10

**Table 6 plants-12-00831-t006:** Correlations (r) among WSOC parameters in the five treatment groups (CK, NPK, BC, NPK + BC, and N + BC).

	WSOC (mg/kg)	C1	C2	C3	FIX	HIX	BIX
**WSOC** **(mg/kg)**	1						
**C1**	−0.089	1					
**C2**	−0.376 **	0.510 **	1				
**C3**	0.319 **	0.409 **	0.020	1			
**FIX**	−0.180	−0.023	−0.366 **	0.475 **	1		
**HIX**	−0.324 **	−0.236	0.432 **	−0.200	−0.359 **	1	
**BIX**	−0.50 *	0.058	−0.205	0.718 **	0.777 **	−0.118	1

WSOC; FIX; HIX; BIX; C1–3 represent the three fluorescent compounds derived from PARAFAC modeling. * *p*-value < 0.05 and ** *p*-value < 0.01.

## Data Availability

Not applicable.

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
