# Peer review of "Effects of Biochar on the Fluorescence Spectra of Water-Soluble Organic Matter in Black Soil Profile after Application for Six Years"

_plants, 2023, doi:10.3390/plants12040831_

Round 1
Reviewer 1 Report
The reference list is, in my opinion, insufficient. I would recommend, e.g., to cite in the Introduction section some of those papers: doi.org/10.1016/j.fuel.2022.126405, doi.org/10.1016/j.ecohyd.2018.11.005, doi.org/10.1016/j.jhazmat.2022.129795, doi.org/10.3390/agronomy11112224, doi.org/10.3390/ma14061335, doi.org/10.3390/agronomy10121903.
Line 69-70: The results of Fan et al… Are they good or not? Please explain to the reader by additional 1-2 sentences in the text.
General remark: you should explain all abbreviations in the manuscript, either in a separate list or in the text where they first appear, like line 75: EEM/PARAFAC – no explanation is given…
Before section 2 (Results): More info on the methodology of measurements needed. I’d recommend to move all the section 3 (Materials and methods) before section 2 (Results).
What are the units for the values in Table 1? Probably [%] but it is not shown.
Fig. 2 should be mentioned in the text before it is shown. Please do so or move the figure.
Table 2: What are 'Em' 'Ex' and other abbreviations - please explain.
Section 2.3: What kind of correlation do the authors mean?
Table 3: What are the units? [%]? Please show them, otherwise the reader has to guess.
Table 4: What are all abbreviations, like FIX, HIX, BIX - please explain. How did you make the correlations? What is 'P'. Explain.
In the Conclusion section please do mention also the results from Figs. 2-3.
Reviewer 2 Report
Dear Authors,
Your manuscript is a valuable contribution with high scientific output. The study of the biochar application is up to date.
I propose you to make corrections. The details are presented herewith below.
Major comments
The manuscript gives an impression of some kind lightweight text, please pay an attention to the justifications, processes and connections of phenomena studied.
What culture(s) did you study?
Many pairs of words in the manuscript are written together, including the data and units of measure (0-60cm).
You use the terms: water-soluble organic matter (WSOM), water-soluble organic carbon (WSOC), soluble organic matter (SOM), soil organic carbon (SOC), and dissolved organic matter (DOM). Was there any need for this diversity? Moreover, the term SOM is already taken for soil organic matter.
Minor comments
The diversity of data on the variants of experiment in the figure 1 and table 1 and in the figure 3 and table 3 is of low significance and is not perceived as proven without statistical data.
Line 154
…according to Fig. 3,
However, the Figure 2 is after this text.
Lines 221-222
…nitrogen and its transformation are the primary essences in regulating the process of humification.
How do your own data support this point of view?
Lines 263-264
However, this region now undergoes severe soil degradation due to excessive land reclamation.
What kind of reclamation do you mean? How does the reclamation degrade the soil? Please clarify.
Line 237
The correlation values in the table 4 are mostly insignificant. However, you did not pay an attention to this fact, nor mention the table 4 in the text above this table.
Lines 266-268
The meteorological data indicated that the long-term average annual rainfall in this area ranged from 486.4 to 543.6 mm.
The precipitation norm is rather low. You applied the studied materials to the upper soil layer. Do you insist that your experiment affected the soil at a depth of 60 cm?
Lines 354-355
(0-20 cm). it demonstrates a positive, cooperative effect between organic-inorganic fertilizer mixture and the topsoil layer mainly due to its aeration condition. Please use the dot at the end of the sentence.
Please do not just state, but explain your results by the processes occurring in the soil as a result of experiment.
399-401
Significant changes in the existing indicators demonstrate that the soil WSOM composition 400 changes significantly after six years of biochar and chemical fertilizer application.
Please give explanations.
Lines 415-416
Nitrogen is the sovereign factor improving synergism effect of chemical fertilizer and biochar
in the humidification process.
Nobody doubt on this matter. Besides, you should give your own original explanations.
The mentioned above flaws can be attributed to other parts of the manuscript.
